# Post-Mortem Cardiac Magnetic Resonance in Explanted Heart of Patients with Sudden Death

**DOI:** 10.3390/ijerph192013395

**Published:** 2022-10-17

**Authors:** Giovanni Donato Aquaro, Benedetta Guidi, Michele Emdin, Angela Pucci, Enrica Chiti, Alessandro Santurro, Matteo Scopetti, Federico Biondi, Aniello Maiese, Emanuela Turillazzi, Giovanni Camastra, Lorenzo Faggioni, Dania Cioni, Vittorio Fineschi, Emanuele Neri, Marco Di Paolo

**Affiliations:** 1Academic Radiology, University of Pisa, 56126 Pisa, Italy; lorenzo.faggioni@unipi.it (L.F.); dania.cioni@unipi.it (D.C.); emanuele.neri@unipi.it (E.N.); 2ASL Toscana Nord-Ovest, 55100 Lucca, Italy; benedetta.guidi@uslnordovest.toscana.it; 3Fondazione Toscana G. Monasterio, 56124 Pisa, Italy; emdin@ftgm.it; 4Scuola Superiore Sant’Anna, 56127 Pisa, Italy; 5Department of Surgical, Clinical and Molecular Pathology and of Critical Area, University of Pisa, 56126 Pisa, Italy; angelapucci@libero.it (A.P.); enricachiti@gmail.com (E.C.); 6Department of Medicine, Surgery and Dentistry-Scuola Medica Salernitana, University of Salerno, 84084 Fisciano, Italy; asanturro@unisa.it; 7Department of Medical Surgical Sciences and Translational Medicine, Sapienza University of Rome, 00189 Rome, Italy; matteo.scopetti@uniroma1.it; 8Cardiology Department, University of Trieste, 34127 Trieste, Italy; biondi.federico@yahoo.it; 9UO Medicina Legale, University of Pisa, 56126 Pisa, Italy; aniello.maiese@unipi.it (A.M.); emanuela.turillazzi@unipi.it (E.T.); marco.dipaolo@unipi.it (M.D.P.); 10Cardiac Department, Vannini Hospital Rome, 00177 Roma, Italy; gcamastra@virgilio.it; 11Department of Anatomical, Histological, Forensic and Orthopaedic Sciences, Sapienza University of Rome, 00185 Rome, Italy; vittorio.fineschi@uniroma1.it

**Keywords:** post-mortem cardiac magnetic resonance, sudden death, sudden cardiac death, autopsy, forensic investigation, histology

## Abstract

Background: We sought to evaluate the diagnostic accuracy of post-mortem cardiac magnetic resonance (PMCMR) of explanted hearts to detect the cardiac causes of sudden death. Methods: PMCMR was performed in formalin-fixed explanted hearts of 115 cases of sudden death. Histological sampling of myocardium was performed using two different approaches: (1) guideline-based sampling; (2) guideline-based plus PMCMR-driven sampling. Results: Forensic diagnosis of cardiac cause of death was ascertained in 72 (63%) patients. When the guideline-driven histological sampling was used, the PMCMR interpretation matched with final forensic diagnosis in 93 out of 115 cases (81%) with sensitivity of 88% (79–95%), specificity of 65% (47–80%), PPV of 84% (78–90%), NPV of 73% (58–84%), accuracy of 81% (72–88%), and AUC of 0.77 (0.68–0.84). When a PMCMR-driven approach was added to the guideline-based one, the matching increased to 102 (89%) cases with a PMCMR sensitivity of 89% (80–94%), a specificity of 86% (67–96%), PPV of 95% (89–98%), NPV of 73% (59–83%), accuracy of 89% (81–93%), and AUC of 0.88 (0.80–0.93). Conclusions: PMCMR has high accuracy to identify the cardiac cause of sudden death and may be considered a valid auxilium for forensic diagnosis. PMCMR could improve histological diagnosis in conditions with focal myocardial involvement or demonstrating signs of myocardial ischemia.

## 1. Introduction

Sudden death is defined as a natural, unexpected, fatal event occurring in an apparently healthy individual within 1 h of symptoms’ onset or, for unwitnessed deaths, as a sudden event occurring in individuals observed alive within 24 h before death. Sudden cardiac death (SCD) is the most frequent cause of sudden death. SCD is a major health problem, accounting for 50% of cardiovascular deaths and 20% of natural deaths in Western countries [1]. Coronary artery disease (CAD) is the most frequent cause of sudden cardiac death in the general population of industrialized countries accounting for 80% of cases in patients aged >50 years. In younger subjects, sudden cardiac death is more often associated with non-ischemic myocardial disease such as hypertrophic cardiomyopathy (HCM), arrhythmogenic cardiomyopathy (ARC), dilated cardiomyopathy, or myocarditis. Other causes of sudden cardiac death in young adults are coronary artery abnormalities and cardiac rhythm disorders caused by channelopathies (e.g., long QT syndrome, short QT syndrome, Brugada syndrome, and catecholaminergic polymorphic ventricular tachycardia), assumption of sympathomimetic drugs, or drugs causing QT prolongation.

Postmortem pathology and forensic medicine with their ancillary examinations (i.e., toxicology, immunohistology, microbiology, chemistry and genetic testing) together with clinical history are regarded as the gold standard to determine the cause of SCD.

Postmortem radiology is increasingly used in forensic practice as a complementary tool to conventional autopsy investigations. Over the last decade, post-mortem cardiac magnetic resonance (PMCMR) imaging was introduced in forensic investigations of natural deaths related to cardiovascular diseases [2,3,4,5,6].

The potentiality of PMCMR was demonstrated in studies evaluating individual cardiac disease such as myocardial infarction, acute ischemia, cardiomyopathies or myocardial toxic damage [7,8,9,10,11].

However, the effectiveness of PMCMR was never evaluated in a systematic study.

The aim of the present study was to evaluate the diagnostic accuracy of PMCMR of explanted hearts to detect cardiac disease in consecutive patients with sudden death and to evaluate its effectiveness in different cardiac conditions.

## 2. Materials and Methods

We performed PMCMR of hearts explanted from autopsies of 115 consecutive subjects, aged 15–80 years, with sudden death which occurred from April 2017 and January 2021 in the northwestern provinces of the Tuscany Region and in the metropolitan area of Rome (Italy).

Autopsy investigations were performed by experienced cardiovascular pathologists in two tertiary institutions: The Department of Surgical, Medical and Molecular Pathology and Critical Medicine of the University of Pisa, Italy and the Department of Anatomical, Histological, Medico-Legal and Locomotor Sciences of the Sapienza University of Rome, Italy.

Conventional autopsy included macroscopic, histologic and toxicology investigations; when appropriate, further laboratory analyses such as chemistry and genetic testing were performed according to the Recommendations on the Harmonization of Medico-Legal Autopsy Rules produced by the Committee of Ministers of the Council of Europe. A whole-body standard autopsy was performed and the heart was explanted in an en bloc removal, according to the current guidelines for autopsy investigation of sudden death [12]. The heart was then excised and fixed in a formalin solution (10% neutral buffer, 4% formaldehyde) and maintained in ambient temperature [11].

After PMCMR was performed, gross transverse sections and histological and ancillary examinations of the heart were carried out.

PMCMR was performed in all the cases with negative extra-cardiac findings at autopsy (potential cardiac death group) and also in cases of individuals with ascertained extracardiac cause of death at whole-body autopsy (non-cardiac death group). PMCMR was performed by investigators blind to the results of the gross autopsy and of clinical data.

Cases of inadequate body preservation (signs of early putrefaction and gas formation at autopsy) at autopsy were ruled out prior to PMCMR to avoid gas-related MR signal interferences.

### 2.1. Post-Mortem Cardiac Magnetic Resonance

PMCMR was carried out with a 1.5 Tesla MRI scanner (General Electric Healthcare^®^, Milwaukee, WI, USA). A dedicated 16-channel cardiac coil and a heart rate simulator with a set heart rate of 60 bpm was present. After the acquisition of triplane conventional localizer images, a 4-chamber localizer view was acquired using a single-shot steady-state free precession sequence (SSFP). Then, a whole-heart 3D-Fat Sat prepared SSFP pulse sequence was acquired with the following parameters: slice thickness of 0.9 mm (1.8 mm interpolated), no gap, FOV 22 × 22 cm, phase FOV 1, matrix 256 × 256, reconstruction matrix 512 × 512, flip angle of 45°, and a TR/TE ratio approximated = 2, slices acquired 184. From the 4-chamber SSFP view, a set of left ventricular short axis views covering both ventricles was acquired using a matrix of 22 × 22 cm and a slice thickness of 8 mm, no gap, and the following pulse sequences:

(1)T1 mapping with modified lock-locker (MOLLI) pulse sequence with a 3(3)3(3)5 setting.(2)T2 mapping using a multi-echo spin-echo (MESE) pulse sequence with 4 different echo times (10.9, 32.8, 54.7, and 76.6 msec).(3)T2* mapping using the multi-echo T2* GRE STARMAP pulse sequence, with 8 different echo times from 2.5 to 16.7 msec.(4)Cine-IR (lock-locker) pulse sequence with 30 images with different inversion times.

Finally, a MAGIC pulse sequence with 22 × 22 cm FOV, 256 × 256 matrix and 4 mm slice thickness with no gap was acquired, with reconstruction of a complete dataset of T1-, T2- and proton density (PD) mapping, T2-weighted fast spin echo (FSE), T1-weighted FSE, PD, Fat Sat-PD and short-tau inversion recovery (STIR) synthetic images. The MR acquisition time of the entire protocol was 25 min.

Use of the image data for the present study was approved by the local ethics committee. PMCMRs were analyzed by blinded MR board-certified investigators, who were unaware of the circumstances of death and of the patient’s medical history.

Image post-processing was performed using a CVi42 (Circle Cardiovascular Imaging Inc^®^, Calgary, AB, Canada) software. All measurement and image interpretation was performed on a picture archiving and communication system (PACS) workstation (ISD7^®^, Sectra, Linköping, Sweden).

Global myocardial T1, T2 and T2* values were measured as the average value of the entire LV myocardium, excluding the regions of myocardium with signal abnormalities in conventional images. Segmental myocardial T1, T2 and T2* were also evaluated. In order to weight myocardial relaxivity properties for the degree of formalin impregnation, myocardial T1-, T2- and T2* were indexed with the same reflexivity value of formalin obtaining, respectively, T1-index, T2-index and T2* index.

Left ventricular mass was measured and regional left ventricular wall thickness (LV WT) was measured as previously reported [13]. LV WT was measured in all 17 myocardial segments on the PMCMR images. The following parameters were also calculated: the average of LV WT, the standard deviation of LV WT, maximal WT, the minimal WT, the difference between the maximal and minimal WT (max-minWT), and LV mass.

As previously validated, a cut-off of >2.4 of the standard deviation of LV WT was used to identify hypertrophic cardiomyopathy [13].

Fat infiltration was identified as intramyocardial “India ink” signs in 3D-SSFP and confirmed using the PD FSE and STIR approach.

A summary of PMCMR algorithms for diagnostic suspicion of cardiac conditions is shown in Table 1.

### 2.2. Forensic Examination of the Heart

According to the guidelines [10], gross examination of the heart involved:

(1)The evaluation of pericardium;(2)The assessment of: coronary arteries’ size, shape, position, number, and patency of the coronary ostia; and the size, course, and “dominance” of the major epicardial arteries by multiple transverse cuts at 3 mm intervals along the course of the main epicardial arteries, including branches such as the diagonal and obtuse marginal;(3)A complete short axis cut of the heart at the mid-ventricular level, and then further parallel transverse slices of ventricles at 1 cm intervals towards the apex. The remainder of right and left ventricles in the basal half of the heart were dissected in the direction of flow of blood.(4)Examination of atrial and ventricular septa, atrio-ventricular valves, ventricular inflows and outflows, and ventricular valves.(5)The wall thickness of mid-cavity free wall (both LV and RV) and of interventricular septum were measured. The transverse dimensions of both ventricles and atria were also measured.

Tissue sampling was performed with two different approaches:

(1)With guideline-oriented sampling;(2)With PMCMR-driven sampling of tissue.

In order to perform histological analysis, 9 blocks of myocardium were collected from the apex, the free wall of the LV (anterior, lateral and posterior), the LV septum (anterior and posterior), and from the free wall of the RV.

Additionally, samples from any area with macroscopic or instrumental alteration were collected.

In the setting of coronary artery disease, relevant atherosclerotic lesions and endoluminal clots were sampled.

Additional PMCMR-driven samples from any area with PMCMR alterations were collected.

All samples were stained with hematoxylin and eosin (H&E). If necessary, histochemical and immunohistochemical staining techniques for the characterization of inflammatory infiltrates or for detecting early ischemic lesion and molecular screening were performed. When needed, histomorphometry analysis served to calculate the extent of fibrofatty replacement in the myocardium. The final diagnosis was defined according to the current guidelines [12].

### 2.3. Statistical Analysis

Diagnostic performance of PMCMR was compared with autopsy. Values are presented as the mean ± standard deviations (SD) or as the median (25th–75th percentiles) for variables with normal and non-normal distributions, respectively. Values with non-normal distribution according to the Kolmogorov-Smirnov test were logarithmically transformed for parametric analysis. Qualitative data are expressed as percentages. Categorical variables were compared by the chi-squared test or the Fisher exact test when appropriate. Continuous variables were compared by the ANOVA t test and analysis of variance or by the Wilcoxon nonparametric test when appropriate. Bonferroni correction was used when needed.

## 3. Results

PMCMR was performed in 115 consecutive cases of sudden death. The mean age at death was 50 ± 19 years (range: 18–81 years). In total, 71 patients were male and 103 patients were white European (7% Asian, 3% African). As shown in Figure 1, 52 deaths occurred at home (or in private places), 34 in a public place (of them, 5 during work activities and 1 during sport activity), and 29 in hospital (during first emergency aids or in emergency rooms). The majority of SD resulted from out-of-hospital cardiac arrests (75%), mostly unwitnessed (61%). Most of the deceased had not-known CAD risk factors (61 without risk factors), despite a median body mass index of 26 kg/m^2^ (IQR 23–31). Sixteen deceased had multiple cardiovascular risk factors; 10 deceased had only one known risk factor for CAD: 5 subjects were active smokers; 2 had type 2 diabetes mellitus; 1 subject had systemic hypertension; 1 had hypercholesterolemia; and 1 had family history of premature CAD. Seven subjects had previous history of ischemic heart disease, and one of valvular heart disease. The median time from death to autopsy was 2 days (IQR 1–4 days).

### 3.1. Guideline-Driven Histological Sampling

The final forensic diagnoses, based on a guideline-driven sampling, identified a cardiac cause of death in 72 cases (63%): ischemic heart disease in 45 cases; ARC in 3; hypertrophic cardiomyopathies in 9; diffuse non-specific myocardial damage in 8 (with history of drug abuse in 5, of sepsis in 2 and of acute idarubicin intoxication in 1); acute myocarditis in 2; and other cardiac conditions in 5 (traumatic heart rupture in 2; cardiac tumor of the interventricular septum in 1; non-specific signs of focal myocardial damage in 2). In 37 (32%) cases, no signs of cardiac involvement were found. Finally, in six patients, advanced decomposition did not allow a complete heart evaluation. At autopsy, the median weight of the heart was 460 g (IQR 362–543); the median heart diameters were 11 cm (10–13) × 12 cm (11–13) × 4 cm (4–5). The median septal wall thickness was 1.5 cm (1.3–1.8). The median LV and RV wall thicknesses were 1.5 cm (1.4–1.8) and 0.5 cm (0.4–0.8), respectively.

The median time from death to PMCMR was 23 days (11–31 days). The median time between autopsy and PMCMR was 21 days (8–29 days). At PMCMR, the median LV myocardial mass was 130 g (103–171). The median septal and LV inferolateral wall thicknesses were both 1.6 cm (1.3–1.9 cm).

PMCMR interpretation matched with the final forensic diagnosis in 93 out of 115 cases (81%): 43 with ischemic heart disease, 3 with ARC, 9 with HCM, 2 with myocarditis, 1 with diffuse myocardial damage (acute idarubicin toxicity), in 1 case with septal neoformation, 2 traumatic heart damage, and 24 cases with non-cardiac death (Figure 2). Yet, PMCMR and forensic diagnosis matched in the two cases of non-specific focal myocardial abnormalities and were both undiagnostic in six because of myocardial signs of advanced post-mortem phenomenon. In contrast, PMCMR was falsely negative in nine cases: it missed ischemic heart disease in two cases and the remaining seven cases of diffuse myocardial damage. Finally, PMCMR was falsely positive in 13 cases of negative forensic diagnosis: suspect of ARC in 9, of ischemic heart disease in 1, and “diffuse myocardial damage” in 3 (eventually resulting in fixation defects with negative cardiac histology).

Overall, PMCMR had a sensitivity of 88% (79–95%), specificity of 65% (47–80%), positive predictive value of 84% (78–90%), negative predictive value of 73% (58–84%), and accuracy of 81% (72–88%), with respect to autoptic diagnosis. The PMCMR area under the curve (AUC) was 0.77 (0.68–0.84).

### 3.2. Guideline-Driven plus PMCMR-Driven Histological Sampling

The PMCMR-driven histologic sampling of the myocardium was performed in addition to the guideline-based approach. With the addition of PMCR-driven sampling, the final forensic diagnosis changed in nine patients. Non-cardiac death decreased to 28 cases. Interestingly, the PMCMR-driven sampling permitted detecting nine more cases of ARC, which would otherwise have gone unnoticed. The final matching between the PMCMR and autopsy was obtained in 102 (89%) cases. PMCMR showed a sensitivity of 89% (80–94%), a specificity of 86% (67–96%), a positive predictive value of 95% (89–98%), a negative predictive value of 73% (58–83%), and an accuracy of 89% (80–94%). The AUC of PMCMR was 0.88 (0.80–0.93).

As evident in Table 2, PMCMR was more effective in detecting cardiac disease in conditions with focal myocardial involvement and was less effective in cases of diffuse myocardial damage.

### 3.3. Ischemic Heart Disease

Ischemic heart disease was diagnosed at autopsy in 45 cases, accounting for 48% of causes of sudden cardiac death. Among them, PMCMR identified ischemic cardiopathy (including myocardial damage and/or obstructive coronary artery disease) in 43 hearts (95%). PMCMR was falsely positive in one case (ectasic right coronary artery with T2w hypointensity in the inferior LV wall suggesting peracute myocardial ischemia), and missed two cases of final forensic diagnosis of ischemic heart disease (it was non-diagnostic in one case, and mis-interpreted as “non-specific” myocardial abnormalities in a case with increased T1 and T2 of the inferior distal wall of LV but missed detecting the obstructive stenosis in a small right coronary artery). An example of PMCMR and histology findings in a case of ischemic heart disease is shown in Figure 3 and Figure 4.

The overall sensitivity of PMCMR in diagnosing ischemic heart disease was 96% (84–99.9%), with a specificity of 98% (92–99.9%), PPV of 97% (85–99.6%), NPV of 97% (89–99%), and accuracy of 97% (93–99%). The AUC of PMCMR for ischemic heart disease was 0.97 (0.93–0.99).

Evaluating coronary artery anatomy, PMCMR missed CAD diagnosis in two cases (6%) and falsely identified significant coronary artery disease in two.

### 3.4. Arrhythmogenic Cardiomyopathy

PMCMR identified 12 cases with myocardial fatty infiltration of right, left or both ventricles. Most ARC had biventricular involvement (eight cases, 66%), followed by left-dominant (two cases, 16%) and right ventricular abnormalities (two cases, 16%). As mentioned above, through guideline-based histologic sampling, the autopsy identified myocardial fibrofatty infiltration only in 3 out of 12 hearts (18%). In contrast, using an additional PMCMR-guided approach to perform histologic sampling in those anatomic regions where MR identified cardiac abnormalities attributable to fatty infiltration, a further nine cases could be diagnosed (fibro-fatty infiltration was found at histology in all these cases). An example of a case of biventricular ARC is shown in Figure 5.

### 3.5. Hypertrophic Cardiomyopathy

PMCMR and forensic diagnoses matched in all nine cases of HCM (an example is in Figure 6). At PMCMR, the LV mass and the average WT of LV myocardial segments were not different between HCM and controls (respectively, 143 ± 33 g vs. 130 ± 39 g, *p* = 0.39 and 17 ± 5 mm vs. 16 ± 3, *p* = 0.54). However, the standard deviation of LV WT of HCM cases diagnosed was significantly higher than in those without (4.6 ± 1.9 vs. 2.1 ± 1, *p* < 0.001). The absolute difference between the maximal and minimal LVWT (max-minD) in HCM cases was 10 ± 5, and in others, 5 ± 2 (*p* < 0.001). The previously reported cut-off of standard deviation of LV WT (>2.4) allowed the identification of all the cases of HCM. Secondary HCM features (such as intramyocardial crypts, coronary artery bridges and apical aneurysm, anomaly of number, or insertion of papillary muscles) were also found in five out of nine cases.

Finally, in seven cases, myocardial signal abnormalities were also found at PMCMR.

### 3.6. Tissue-Mapping Analysis

The mean relaxation times for formalin were: T1, 1975 ± 181 ms; T2, 560 ± 73 ms; and T2* was 324 ± 84 ms. In patients with non-cardiac death, the mean myocardial relaxation times were: T1, 430 ± 24 ms; T2, 59 ± 4 ms; and T2*, 43 ± 5. The myocardial relaxation times had some changes within the first week after formalin fixation (a decrease in T1, T2 and T2*), then remained stable in the time window between the first week and the first 60 days after formalin fixation. In patients with cardiac death, abnormalities of regional myocardial T1 and T2 were consistent with the signal abnormalities of the T1-weighted and T2-weighted pulse sequence.

The average values of myocardial T1, T2 and T2* for different cardiac conditions are shown in Table 3. Interestingly, the patients with a final diagnosis of ARC (guideline + PMCMR-guided) had a lower average myocardial T1 compared to the whole population, to ischemic heart disease, and to non-cardiac death. The T1 index of ARC was significantly lower than that of all the other groups.

## 4. Discussion

In this study, the result of PMCMR of explanted hearts was compared to the standard forensic procedures for the evaluation of cardiac diseases in a cohort of cases of sudden death. The histological evaluation of the forensic procedure was performed following different approaches:

(1)Performing guideline-oriented sampling of myocardial tissue;(2)Performing PMCMR-driven tissue sampling.

PMCMR demonstrated a good matching with the standard forensic procedure with the guideline-oriented approach (matching in 80% of cases), which increased by adding the PMCMR-driven approach (matching in 88% of cases). Interestingly, the PMCMR-driven approach allowed performing a further nine diagnoses of ARC that would have been missed using the guideline-based approach.

PMCMR demonstrated 88% sensitivity and 65% specificity in detecting cardiac disease in this cohort of sudden death. Applying also the PMCMR-driven approach, sensitivity remained almost unchanged (89%) but specificity increased to 86%. PMCMR was particularly effective in the diagnosis of conditions with focal myocardial abnormalities such as cardiomyopathy (ARC and HCM) or ischemic heart disease. In contrast, PMCMR failed to identify most of the cases with diffuse myocardial involvement because the imaging aspect of such conditions, mostly involving the mid-wall layer, could mimic fixation defects.

In living patients, CMR represents the gold-standard imaging tool to evaluate cardiac morphology and function and to characterize tissues through the assessment of myocardial magnetic properties and the use of gadolinium-based contrast-enhanced sequences. CMR is considered the gold-standard imaging technique to diagnose myocarditis, and cardiomyopathy such as HCM and dilated cardiomyopathy. It is also able to distinguish between ischemic and non-ischemic myocardial disease and to detect signs of myocardial damage such as fat infiltration, edema and myocardial fibrosis. Compared to CMR, PMCMR has several advantages: the high image quality and spatial resolution, resulting from the absence of cardiac motion or breathing-related artefacts, as well as the absence of scan time limitations and of specific absorption rate restrictions. On the other hand, PMCMR cannot rely on contrast enhancement pulse sequences and may be negatively affected by post-mortem phenomena of the deceased bodies, such as autolysis/putrefaction, cooling or rigor mortis. Two different approaches of PMCMR are currently used: (a) whole-body and (b) explanted-heart PMCMR. Whole-body PMCMR has the main advantage of allowing the contemporary evaluation of other organs but it is more susceptible to post-mortem phenomena, to the temperature-dependent changes in myocardial T1 and T2 passing from cooling to an ambient temperature, and to practical issues correlated with the complexity of a procedure, including logistic problems such as transportation of bodies, time constraints, and disturbance to routine clinical casework [5]. PMCMR of an explanted heart may overcome these limitations because the fixation of the heart in formaldehyde immediately after explantation permits conserving the organ at an ambient temperature, saving it from post-mortal alteration progress and improving imaging resolution (using a very low field of view).

In our population, with male preponderance and a mean age of 50 years, ischemic heart disease was found as the cause of almost half of sudden cardiac death cases. The diagnosis of ischemic heart disease was made by the identification of obstructive CAD and/or of ischemic myocardial damage. Overall, PMCMR was able to identify 95% of cases with ischemic heart disease. The mismatch between PMCMR and the final forensic diagnosis referred mostly to the identification of obstructive CAD: PMCMR was falsely positive in two cases, as well as missing CAD in two. The spontaneous formation of post-mortem thrombi, the presence of air bubbles, or the extrinsic compression of the coronary lumen could explain the false-positive results of PMCMR, whereas small vessel diameters could be the cause of false-negative responses.

However, the advantage of PMCMR in ischemic heart disease is the capability to identify early signs of myocardial ischemia [8]. Indeed, as previously demonstrated [2,7], PMCMR is able to detect signs of myocardial ischemia in the peracute phase following coronary occlusion in T2 images as a core of signal hypointensity (or low T2) surrounded by a hyperintense border (high T2). In a recent study, PMCMR was performed in explanted hearts of pigs with arrhythmic death caused by coronary occlusion: a group with death within 40 min of occlusion and a group with death occurring between 40 and 90 min of occlusion [8,9]. In that study, PMCMR was compared with the immunohistochemical evaluation in cardiomyocytes of the altered proportion and redistribution of phosphorylated versus non-phosphorylated connexin 43 which is an established molecular marker of myocardial ischemia. From the results, PMCMR was able to detect early signs of myocardial ischemia even those occurring before 90 min of coronary occlusion. This finding was relevant because the standard forensic procedures are unable to detect myocardial signs of ischemia occurring within 3 h of coronary occlusion. In these circumstances, only the identification of obstructive CAD could permit identifying ischemic heart disease as the cause of death. PMCMR showed a very high diagnostic performance both in detecting ischemic cardiomyopathy (tissue characterization) and coronary artery disease (vessel anatomy), reflecting its notable potential to match ischemic tissue injuries with the concomitant disease of the culprit coronary artery.

One of the most important findings of the present study was the demonstration of the role of PMCMR in the diagnosis of ARC. Overall, ARC was eventually diagnosed in 12 patients of this population, but in only 3 of them, the final diagnosis was achieved by using only the guideline-oriented sampling of myocardial tissue. Myocardial sampling driven by PMCMR was able to detect signs of fat infiltration in nine more patients. The explanation of this result is linked to the high spatial resolution of PMCMR and to its capability to evaluate soft-tissue abnormalities and, particularly, fat infiltration. For this purpose, a 3D fat saturation SSFP was used, resulting in the acquisition of 184 slices of 0.9 mm thickness, without gap, and a final voxel size of 0.9 × 0.9 × 0.9 mm. In these images, regions of fat infiltration were detected as well-defined hypointense regions within apparently normal myocardium. In contrast, for histology, ventricles were cut using 1cm-thickness short axis slices and blocks of myocardium were sampled as indicated in the current guidelines for histologic analysis. Indeed, this great difference in “spatial resolution” might explain the high effectiveness of PMCMR in detecting the region of intramyocardial fat infiltration as well as the significant improvement in the PMCMR-driven approach over guideline-oriented tissue sampling. The present study was conducted in Italy where ARC was the first non-ischemic cause of sudden cardiac death [14], particularly in young patients. This may explain the great prevalence of ARC in our population (10%). However, in most of these patients, the final diagnosis was obtained only with the guidance of PMCMR.

PMCMR was also able to detect HCM as the cause of death using both morphological and tissue features. Despite the rigor mortis condition, the identification of asymmetrical hypertrophy could be made by evaluating the standard deviation of wall thickness that was significantly higher in HCM than in other conditions [13]. Secondary phenotypic features, such as intramyocardial coronary artery bridge, multiple myocardial crypts, apical aneurysms with thinned walls, hypertrophy and anomalies of number, and/or insertion of papillary muscles, may help to make a diagnosis of this cardiomyopathy. Finally, myocardial signal abnormalities such as increased T1 and hyperintensity in T2-weighted images in the hypertrophic walls, as a marker of acute myocardial damage [15], could be useful for diagnosis. Differential diagnosis among HCM, cardiac amyloidosis and Fabry disease could be performed in vivo: in HCM, hypertrophy is mostly asymmetrical and signal abnormalities are focal and located in hypertrophic segments; in cardiac amyloidosis. pseudo-hypertrophy is concentric and signal abnormalities (increased T1) are diffuse; and finally, in Fabry disease, pseudo-hypertrophy is mostly concentric, but myocardial T1 is diffusely decreased because of an intracellular overload of sphingolipids. However, our population did not include a case of cardiac amyloidosis or Fabry disease, and then we could not indagate the effectiveness of PMCMR to perform this differential diagnosis in explanted hearts.

From the technical point of view, in this large cohort of explanted hearts, we found that myocardial T1, T2 and T2* remained relatively stable over time, after an initial drop-out occurring in the first 7 days following formalin fixation. A recent study by Ebata and coworkers found the same pattern of myocardial relaxation times [16]. These findings suggest that PMCMR in explanted hearts should be performed after 1 week of fixation and that it can be performed with the same results also after months.

Noteworthily, myocardial T1 and T1-index (the ratio between myocardial T1 and formalin T1) were lower in ARC than in other conditions. This lower T1 could be explained by the presence of intramyocardial fat that decreases T1 or by the increased permeability of formalin in the interstitial spaces because of desmosomial abnormalities of the gap junction. However, these findings need to be confirmed by further evidence.

Some study limitations should be mentioned. First, our population did not include all the cardiac conditions that potentially could cause SCD. However, this was a real-life study and the distribution of cardiac disease found reflects the prevalence of the causes of SCD in Italy (in order of prevalence: ischemic heart disease, ARC and HCM).

Second, inadequate body preservation with signs of early myocardial putrefaction, not permitting the identification of the cause of death, was found in a not-negligible percentage of cases. This may be explained by the fact that in 45% of cases, death occurred at home and we decided to include consecutive cases without excluding those with a possible long time gap between death and autopsy.

## 5. Conclusions

This study demonstrated that PMCMR has high accuracy to identify the cardiac cause of sudden death and may be considered a valid auxilium for forensic diagnosis. PMCMR could improve histological diagnosis in conditions with focal myocardial involvement (as in ARC) or to demonstrate signs of myocardial ischemia when death occurs in the peracute or acute phase. In contrast, PMCMR is less effective in conditions causing diffuse myocardial damage. PMCMR of an explanted heart is useful for the detection of cardiac alterations that may be missed by visual examination and random microscopic evaluation, and can be proposed as a complementary guide for histological analysis in sudden death cases, without evident cardiac macroscopic abnormalities.

## Figures and Tables

**Figure 1 ijerph-19-13395-f001:**
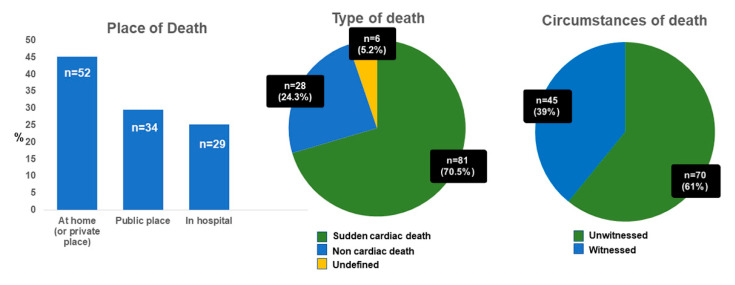
Circumstances of death. As evident by the bar-plot (leftmost graph), most of the deaths occurred at home or in public places, 70% were cardiac deaths and 61% occurred in the absence of a witness.

**Figure 2 ijerph-19-13395-f002:**
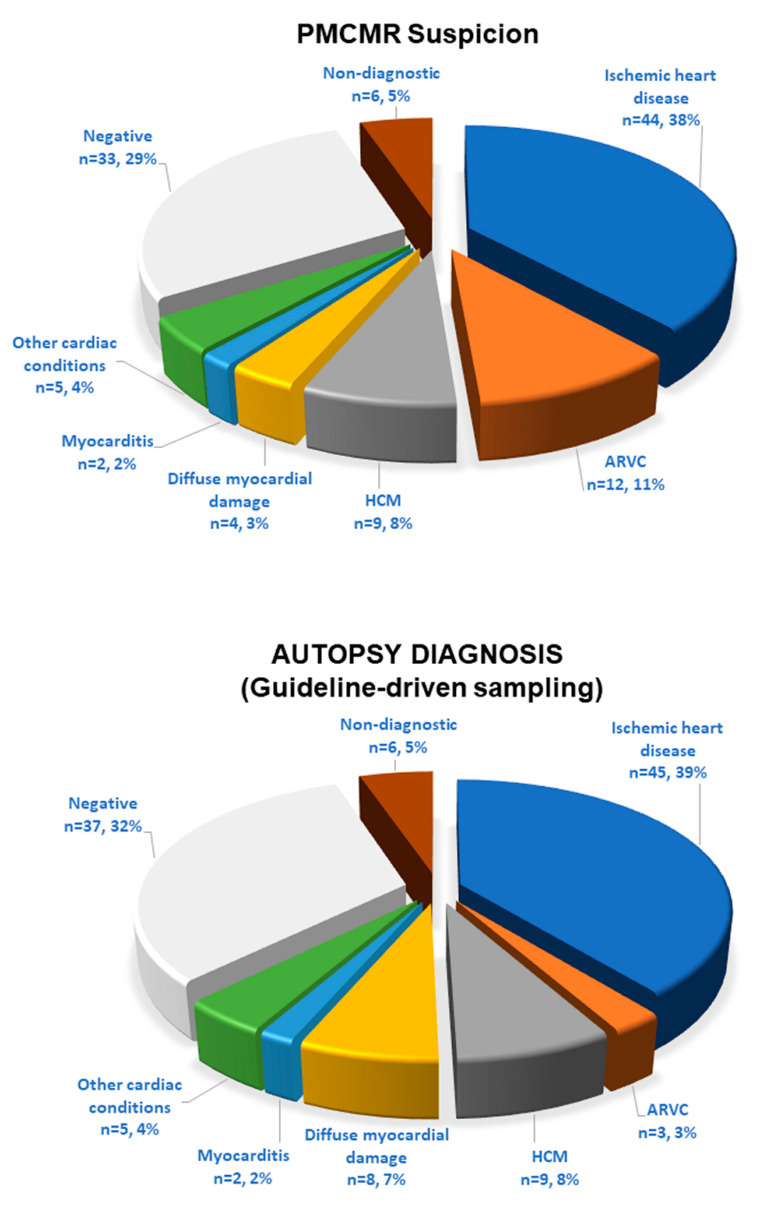
Pie charts of PMCMR suspicion and histological diagnosis. The upper graph shows the initial diagnostic suspicion given by the PMCMR. The lower graph shows the histological diagnosis achieved following the guideline for tissue sampling.

**Figure 3 ijerph-19-13395-f003:**
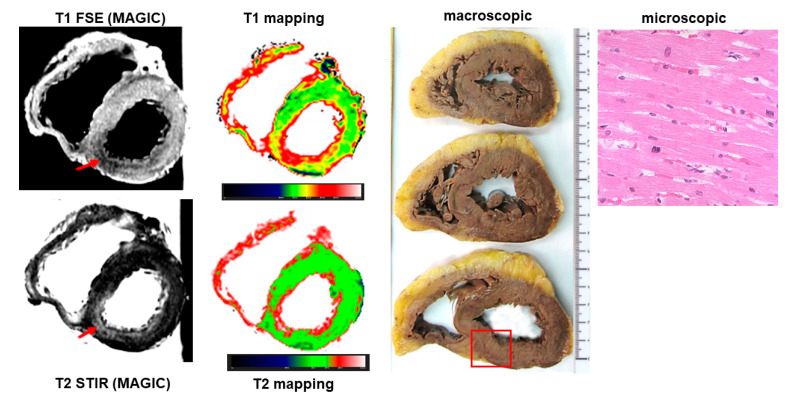
PMCMR and histology in a case of ischemic heart disease. The T1- and T2-weighted images showed an area of, respectively, hypo- and hyperintensity in the inferior wall of the left ventricle. In the same region, mapping techniques demonstrated increased myocardial T1 and T2. The pattern of distribution and kind of signal abnormalities, together with the evidence of significant stenosis of the right coronary artery (see Figure 4), raised the suspicion of death for ischemic heart disease. Interestingly, gross and microscopic histology did not show any abnormality in the inferior wall.

**Figure 4 ijerph-19-13395-f004:**
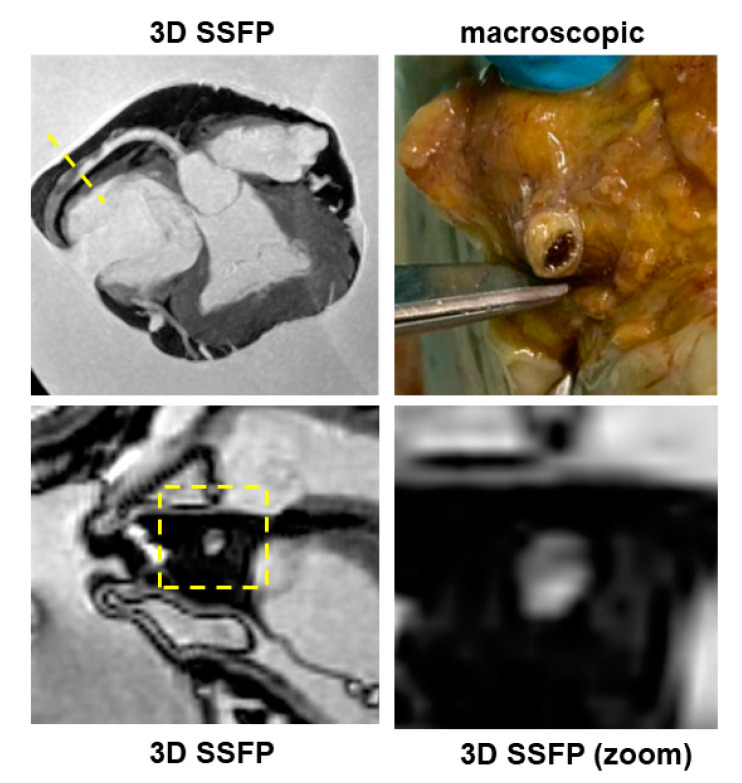
Coronary anatomy by PMCMR and histology. These images are of the same case as Figure 3. A significant stenosis of the right coronary artery was found both in PMCMR and at gross histology.

**Figure 5 ijerph-19-13395-f005:**
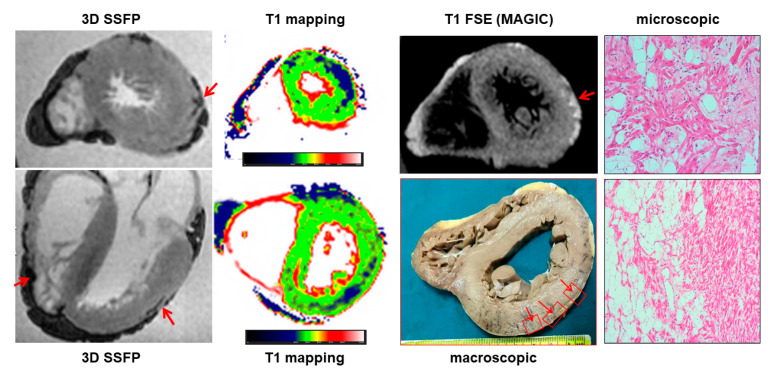
A case of arrhythmogenic cardiomyopathy. In this case, the 3D-SSFP images show an “india ink” artefact in the middle lateral wall of the left ventricle and in the lateral free wall of the right ventricle (arrows), suggesting fat infiltration. At T1 mapping, a large area of decreased T1 was found in the lateral wall of the left ventricle. The PMCMR-driven tissue sampling permitted identifying islands of fat infiltration at histology.

**Figure 6 ijerph-19-13395-f006:**
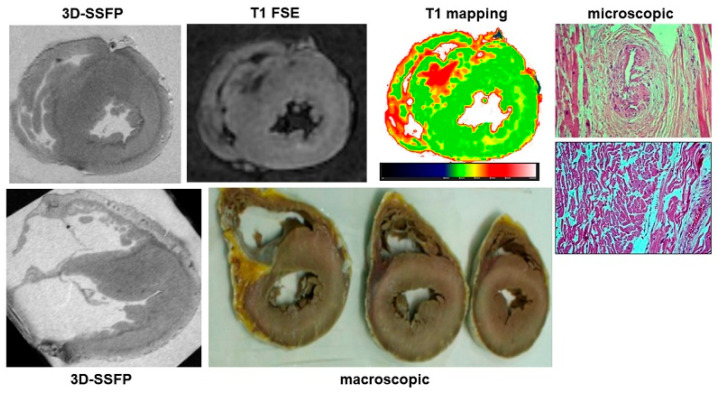
A case of hypertrophic cardiomyopathy. In 3D-SSFP sections, the asymmetrical hypertrophy of the interventricular septum is evident. In the same region, a large area of hypointensity in T1-FSE and increased T1 at mapping were found. Finally, myocardial disarray and perivascular and interstitial fibrosis were found at histology.

**Table 1 ijerph-19-13395-t001:** PMCMR features in cardiac diseases.

Cardiac Disease Suspicion	PMCMR Features
**Ischemic Heart Disease:**	
Coronary artery disease	Identification of obstructive coronary artery stenosis with or without thrombi
Ischemic myocardial damage	Signs of myocardial damages in territory of involved coronary artery. Peracute phase: T2 hypointense core (low T2, low T1) surrounded by hyperintense border (high T2, high T1). Acute phase: T2 hyperintense, T1 hypointense area (high T2, high T1). Chronic phase: slight T2 and T1 hypointensity (low T2 and high T1), with consistent wall thinning. Possibly fat metaplasia within the area. Hemorrhagic infarction: Low T2* + thickened wall.
Hypertrophic cardiomyopathy	- Heterogeneity of wall thickness (SD > 2.4, measured in 17 wall segments) with or without increase in LV mass index. - Secondary features: papillary muscle abnormalities (number, morphology or insertion), myocardial crypts, and coronary artery bridge. - Focal myocardial damage: region of increased T1/T2 in segments with hypertrophy.
Arrhythmogenic cardiomyopathy	- Areas of intramyocardial fat infiltration of RV, LV or biventricular (“india ink” signs in 3D SSFP, hyperintesity in PD FSE, and hypointensity in STIR). - Focal or diffuse thinning of RV walls.
Acute myocarditis	Sub-epicardial (or less frequently, mid-wall) areas of hyperintensity in T2 (and/or increased T2 and T1).
Diffuse myocardial damage	Diffuse myocardial abnormalities (high or low T1, high or low T2), in the absence of hypertrophy (normal LV mass).
Aspecific findings	Focal areas of signal abnormalities (areas of focal increase in T1 or T2) with absence of any supportive criteria for any other conditions.

**Table 2 ijerph-19-13395-t002:** Effectiveness of PMCMR in identifying cardiac disease in the principal diagnosis.

Kerrypnx	Forensic Diagnosis	PMCMR							
		TP	FP	TN	FN	Sensitivity	Specificity	Accuracy	AUC
Ischemic heart disease	45	43	1	24	2	96 (85–99)	96 (80–99)	96 (88–99)	0.96 (0.88–0.99)
Arrhythmogenic cardiomyopathy (guideline)	3	3	9	24	0	100 (29–100)	73 (55–87)	75 (58–88)	0.87 (0.7–0.96)
Arrhythmogenic cardiomyopathy (guideline + PMCR)	12	12	0	24	0	100 (73–100)	100 (85–100)	100 (90–100)	1 (0.9–1)
Hypertrophic cardiomyopathy	9	9	0	24	0	100 (66–100)	100 (86–100)	100 (89–100)	1 (0.9–1)
Diffuse myocardial disease	8	1	3	24	7	13 (2–53)	88 (72–97)	71 (54–85)	0.51 (0.33–0.68)

**Table 3 ijerph-19-13395-t003:** T1, T2 and T2* mapping in cardiac conditions (*, sign. vs. other conditions, *p* < 0.05).

	Whole Population	Ischemic Heart Disease	Arrhythmogenic Cardiomyopathy	Hypertrophic Cardiomyopathy	Diffuse Myocardial Injury	Non-Cardiac Death	*p* Value
	n = 115	n = 45	n = 12	n = 9	n = 8	n = 28	
Myocardial T1 (ms)	438 ± 64	461 ± 89	383 ± 56 *	410 ± 48	412 ± 67	430 ± 24	0.09
Myocardial T2 (ms)	59 ± 7	60 ± 9	59 ± 5	61 ± 9	60 ± 3	59 ± 4	0.99
Myocardial T2* (ms)	43 ± 7	43 ± 7	46 ± 3	48 ± 8	38 ± 5	43 ± 5	0.21
T1 index%	22 ± 4	23.7 ± 4	17.8 ± 3 *	21.6 ± 4	23.4 ± 6	21 ± 4	0.014
T2 index%	11 ± 3	11.7 ± 4	10 ± 2	9.8 ± 3	13.5 ± 5	11 ± 2	0.12
T2* index%	20 ± 6	22.3 ± 6	16 ± 5	15 ± 4	16 ± 4	16 ± 4	0.96

## Data Availability

Datasets are available on request by contacting the corresponding author.

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
