# Peer review of "Post-Mortem Cardiac Magnetic Resonance in Explanted Heart of Patients with Sudden Death"

_ijerph, 2022, doi:10.3390/ijerph192013395_

Round 1

Reviewer 1 Report

The authors present a systematic study on PMCMR compared to conventional forensic methods in the diagnosis of SCD. The paper is overall well written, concise,  and well structured.

The presentation of methods and results is comprehensive; however, the figures lack clarity: the absolute numbers given in figure 1/"type of death" do not match the ones in the results section; and in figure 2, the legend mentions a "lower right graph" that is not shown.

The discussion section is sufficient.

The manuscript shows some obvious spelling and grammar mistakes, and should be thoroughly checked.

Author Response

Dear Editor,
Thank you very much for your kind email regarding our manuscript 
We appreciate the thoughtful comments and questions provided by the reviewers, and have revised the manuscript accordingly. We also have answered all reviewer questions, point by point, in this response letter, and cited where these changes are located in the revised manuscript.
We hope this submission will meet with the approval of the reviewers and editor. We look forward to your feedback on this revised manuscript.
Again, we thank the editorial staff and reviewers for their care and time taken in consideration of this manuscript.

Answers to Reviewers

Rewiever 1#: The authors present a systematic study on PMCMR compared to conventional forensic methods in the diagnosis of SCD. The paper is overall well written, concise,  and well structured.

Answer: many thanks 

Rewiever 1#: The presentation of methods and results is comprehensive; however, the figures lack clarity: the absolute numbers given in figure 1/"type of death" do not match the ones in the results section; and in figure 2, the legend mentions a "lower right graph" that is not shown.

Answer: thanks for this comment. Legends of figure 1 and 2 were uncorrect. we fixed these errors. In legend of figure 1 "70% were cardiac deaths and 61% occurred in absence of witness". The color scheme of the "circumstances of death" of figure 1 was inverted. We fixed errors of legend of figure 2: "Pie-charts of PMCMR suspicion and histological diagnosis. The upper graph shows the initial di-agnostic suspicion given by the PMCMR. The lower graph the histological diagnosis achieved fol-lowing the guideline for tissue sampling" 

Rewiever 1#:The discussion section is sufficient. The manuscript shows some obvious spelling and grammar mistakes, and should be thoroughly checked.

Answer: we checked and fixed erros. Thanks

Reviewer 2 Report

Authors studied the relevance of post mortem MRI of hearts sampled at autopsy and formolin-fixed to diagnose causes of sudden cardiac death. The design of the study is interesting (comparison of histology/MRI diagnostics), and sudden cardiac death is a major health issue. Indeed, as the authors hightlighted in the introduction, the cause of sudden cardiac death is rarely obtained due to the low number of autopsies. Even when an autopsy is performed, histology examination of the heart should be performed by a experienced cardio-pathologist (Guidelines for autopsy investigation of sudden cardiac death:2017 update from the association for european cardiovascular pathology. Virchows Archiv).  However, some pathologies required extensive sampling of the heart to be made (arrhythmogenic cardiomyopathy...).  Post mortem MRI could be of great help to perform driven-sampling of the heart in such cases. 

Figures are clear and MRI versus macroscopic or histology pictures are well chosen. 

- Does th authors know if the hearts were cleared from any post mortem blood before fixation and MRI ? which could explain the false diagnosis iof CAD by MRI approach. 

- As recommanded by Basso et al. guidelines, the number of samples performed for histology could be precised, since it is a good indicator of the quality of the examination. 

- line 214 : since early putrefaction was an exclusion criteria, please explained the examination of 6 hearts with "advanced decomposition" ? 

- line 210 : What is the definition of "Diffuse myocardial disease" or "damage" ? This diagnosis seems unspecific to explain a sudden cardiac death, as it encompasses "drug abuse, sepsis and idarubicin intoxication". Drug abuse, sepsis and such intoxication are potential causes of death, ruling out sudden cardiac death. Therefore, the comparison with MRI diagnosis could be biased. Plus what is the difference with "non-specific myocardial damage" line 212 ?

- The rate of biventricular AC is striking, since the right ventricular form is the most frequent in most studies. However, MRI cannot differentiate between fatty and fatty-fibrosis infiltrations, and fatty infiltration alone is quite a frequent finding, notably in obese women.  Did the histology examination confirm a true fatty-fibrosis infiltrate in all cases ?  Does the authors have some information about medical history or genetic testing in the family ? 

Minor points :

- 1 author lacks an affiliation

- Figures are not well referenced in the main text, please add the reference to Figure 1 and 2 notably.  

Overall, I thank the authors for the remarkable illustrations, well-described histology and MRI procedures, and their work in sudden cardiac death. Another study about genetic testing and follow-up of the relatives in the cases of AC detected by post mortem MRI could be a very good future contribution.  

Author Response

Dear Editor,
Thank you very much for your kind email regarding our manuscript 
We appreciate the thoughtful comments and questions provided by the reviewers, and have revised the manuscript accordingly. We also have answered all reviewer questions, point by point, in this response letter, and cited where these changes are located in the revised manuscript.
We hope this submission will meet with the approval of the reviewers and editor. We look forward to your feedback on this revised manuscript.
Again, we thank the editorial staff and reviewers for their care and time taken in consideration of this manuscript.

Answers to Reviewers

Reviewer 2#: Authors studied the relevance of post mortem MRI of hearts sampled at autopsy and formalin-fixed to diagnose causes of sudden cardiac death. The design of the study is interesting (comparison of histology/MRI diagnostics), and sudden cardiac death is a major health issue. Indeed, as the authors hightlighted in the introduction, the cause of sudden cardiac death is rarely obtained due to the low number of autopsies. Even when an autopsy is performed, histology examination of the heart should be performed by a experienced cardio-pathologist (Guidelines for autopsy investigation of sudden cardiac death:2017 update from the association for european cardiovascular pathology. Virchows Archiv).  However, some pathologies required extensive sampling of the heart to be made (arrhythmogenic cardiomyopathy...).  Post mortem MRI could be of great help to perform driven-sampling of the heart in such cases. 

Answer: many thanks for the recognition of the value of our study

Reviewer 2#: Figures are clear and MRI versus macroscopic or histology pictures are well chosen. 

Answer: thanks

Reviewer 2#: - Does the authors know if the hearts were cleared from any post mortem blood before fixation and MRI ? which could explain the false diagnosis iof CAD by MRI approach. 

Answer: Hearts were washed before formalin fixation, however some air bubble or post-mortem thrombi may have remained within coronary artery. Pathologist tried to keep the integrity of the heart, including coronary artery, before MRI.  

Reviewer 2#: - As recommanded by Basso et al. guidelines, the number of samples performed for histology could be precised, since it is a good indicator of the quality of the examination. 

Answer: we agree with the reviewer. Our pathologists followed current guidelines: 9 samples of ventricular myocardium + 2 for atrii. Then, additional sampling were made driven by gross examination or by MRI. We included the number of samples in the method section. 

Reviewer 2#: - line 214 : since early putrefaction was an exclusion criteria, please explained the examination of 6 hearts with "advanced decomposition" ? 

Answer: early putrefaction was an exclusion criteria when detected at autopsy. In fact we excluded those cases. However, some myocardial post-mortem phenomenon could not be seen macroscopically before cutting the heart. So, sometimes we performed MRI and found myocardial post-mortem abnormalities during the examinations and then it was confirmed with the gross histology.  

Reviewer 2#: - line 210 : What is the definition of "Diffuse myocardial disease" or "damage" ? This diagnosis seems unspecific to explain a sudden cardiac death, as it encompasses "drug abuse, sepsis and idarubicin intoxication". Drug abuse, sepsis and such intoxication are potential causes of death, ruling out sudden cardiac death. Therefore, the comparison with MRI diagnosis could be biased. Plus what is the difference with "non-specific myocardial damage" line 212 ?

Answer: We agree that this point was not well explained. We distinguished between regional damage and diffuse myocardial damage. Regional could be secondary to an infarction or a regional scar for cardiomyopathies etc etc. Diffuse damage means that the signs of myocardial damages were found in all the LV sample. We compared MRI (blinded of clinical data) with autopsy and with all the weapons of autopsy, including anamnesis. So, At MRI a diffuse abnormality of myocardial T1, T2 could be evaluated, and it was interpreted as a diffuse myocardial damage. At autopsy the diagnosis was completed. At line 212 "non-specific myocardial damage" is different from "Diffuse myocardial disease" because it was focal, small, and even shaded and not specific for some cardiac condition.  We changed the sentence with "a diffuse non-specific myocardial damage in 8 (with history of drug abuse in 5, of sep-sis in 2 and of acute idarubicin intoxication in 1)" and "non-specific signs of focal myocardial damage in 2".

Reviewer 2#: - The rate of biventricular AC is striking, since the right ventricular form is the most frequent in most studies. However, MRI cannot differentiate between fatty and fatty-fibrosis infiltrations, and fatty infiltration alone is quite a frequent finding, notably in obese women.  

Answer: We agree that the percentage of biventricular AC  is high. But, these data should be evaluated considering that with histology alone only 3 pts had a diagnosis of AC (1 biv and 2 lone RV), and this is concordant with literature. In a recent in-vivo studies with cardiac-Magnetic resonace. For instance in a recent  multicenter study (aquaro et al JACC 2020), among 140 pts with definite diagnosis of AC, biventricualr were 34%. The % changed when the diagnosi was made driven by CMR (from 2 to 12 pts) with greater prevalence of biventricular aC.  In the above study, as well in that of Smith (circulation 2020), in AC LV involvement was associated with greater risk of malignant arrythmic events as sudden death or cardiac arrest. Then, should not surprise that many pts with AC and sudden death have LV involvement (biventricular or LV dominant). The percentage is greater than previous studies with pathology alone, and this underlines the diagnostic role of MRI that allows detection of small areas of fatty infiltration that could have been missed without MRI. The prevalence of biventricular AC or LV dominant AC in autopsy studies should be re-evaluated by the combination MRI+histology. 

Reviewer 2#: Did the histology examination confirm a true fatty-fibrosis infiltrate in all cases ?  

Answer: We specify this point: "fibro-fatty infiltration was found at histology in all these cases". 

Reviewer 2#: Does the authors have some information about medical history or genetic testing in the family ? 

Answer: all the 12 pts with ARC have not family history of ARVC or genetic assay. After the diagnosis, some family members underwent genetic assay (on clinical indications). the clinical evaluation of family members was beyond the aim of this study, however, in future studies we could collect this data.

Minor points :

Reviewer 2#:  - 1 author lacks an affiliation
Answer: we fixed this, thanks.

Reviewer 2#: - Figures are not well referenced in the main text, please add the reference to Figure 1 and 2 notably.  

Answer: we corrected the references. 

Reviewer 2#: Overall, I thank the authors for the remarkable illustrations, well-described histology and MRI procedures, and their work in sudden cardiac death. Another study about genetic testing and follow-up of the relatives in the cases of AC detected by post mortem MRI could be a very good future contribution.  

Answer: many thanks for this suggestion. For AC, we are performing genetic evaluation in some family members (based on clinical decision).